# Isolation, Antibacterial Activity and Molecular Identification of Avocado Rhizosphere Actinobacteria as Potential Biocontrol Agents of *Xanthomonas* sp.

**DOI:** 10.3390/microorganisms12112199

**Published:** 2024-10-31

**Authors:** Jesús Rafael Trinidad-Cruz, Gabriel Rincón-Enríquez, Zahaed Evangelista-Martínez, Luis López-Pérez, Evangelina Esmeralda Quiñones-Aguilar

**Affiliations:** 1Laboratorio de Fitopatología, Unidad de Biotecnología Vegetal, Centro de Investigación y Asistencia en Tecnología y Diseño del Estado de Jalisco A.C. (CIATEJ), Camino Arenero 1227, El Bajío del Arenal, Zapopan 45019, Jalisco, Mexico; jesus.trinidadcruz@gmail.com; 2Currently Independent Researcher, Zapopan 45019, Jalisco, Mexico; 3Centro de Investigación y Asistencia en Tecnología y Diseño del Estado de Jalisco A.C. Subsede Sureste, Tablaje Catastral 31264 Km 5.5 Carretera Sierra Papacal—Chuburná Puerto, Parque Científico y Tecnológico de Yucatán, Mérida 97302, Yucatán, Mexico; zevangelista@ciatej.mx; 4Instituto de Investigaciones Agropecuarias y Forestales, Universidad Michoacana de San Nicolás de Hidalgo, Km 9.5 Carr. Morelia Zinapécuaro, Tarímbaro 58880, Michoacán, Mexico; luis.lopez.perez@umich.mx

**Keywords:** bacterial spot, *Streptomyces*, *Amycolatopsis*, biological control, microwave irradiation, antagonist, antibiosis

## Abstract

Actinobacteria, especially the genus *Streptomyces*, have been shown to be potential biocontrol agents for phytopathogenic bacteria. Bacteria spot disease caused by *Xanthomonas* spp. may severely affect chili pepper (*Capsicum annuum*) crops with a subsequent decrease in productivity. Therefore, the objective of the study was to isolate rhizospheric actinobacteria from soil samples treated by physical methods and evaluate the inhibitory activity of the isolates over *Xanthomonas*. Initially, soil samples collected from avocado tree orchards were treated by dry heat air and microwave irradiation; thereafter, isolation was implemented. Then, antibacterial activity (AA) of isolates was evaluated by the double-layer agar method. Furthermore, the positive/negative effect on AA for selected isolates was evaluated on three culture media (potato-dextrose agar, PDA; yeast malt extract agar, YME; and oat agar, OA). Isolates were identified by 16S rRNA sequence analysis. A total of 198 isolates were obtained; 76 (series BVEZ) correspond to samples treated by dry heat and 122 strains (series BVEZMW) were isolated from samples irradiated with microwaves. A total of 19 dry heat and 25 microwave-irradiated isolates showed AA with inhibition zones (IZ, diameter in mm) ranging from 12.7 to 82.3 mm and from 11.4 to 55.4 mm, respectively. An increment for the AA was registered for isolates cultured on PDA and YME, with an IZ from 21.1 to 80.2 mm and 14.1 to 69.6 mm, respectively. A lower AA was detected when isolates were cultured on OA media (15.0 to 38.1 mm). Based on the 16S rRNA gene sequencing analysis, the actinobacteria belong to the *Streptomyces* (6) and *Amycolatopsis* (2) genera. Therefore, the study showed that microwave irradiation is a suitable method to increase the isolation of soil bacteria with AA against *Xanthomonas* sp. In addition, *Streptomyces* sp. BVEZ 50 was the isolate with the highest IZ (80.2 mm).

## 1. Introduction

Worldwide, phytopathogenic species of the genus *Xanthomonas* are a serious threat for agriculture production. Some species cause serious diseases in a wide range of economically important crops, affecting their productivity [1]. For instance, the bacterial spot on chili pepper (*Capsicum annuum* L.) crops caused by *Xanthomonas* spp. is a disease which decreases the fruit production by up to 30%, and depending on the severity of the disease, the losses could be even more dramatic [2,3]. The control of the disease by commercial formulations based on copper has been ineffective due to its excessive use, causing the selection of copper-resistant *Xanthomonas* strains [4,5]. Antagonistic microorganisms of the genera *Bacillus*, *Pseudomonas*, and *Streptomyces*, isolated from soil, rhizosphere, phyllosphere, and plant endophytes, could be a sustainable alternative or a complement to an integrated management program to control bacterial diseases in plants caused by *Xanthomonas* spp. [6].

Microbial antagonists associated with avocado rhizosphere—mainly belonging to the genus *Bacillus*—have demonstrated to be potential biocontrol agents against phytopathogenic fungi (*Fusarium* spp., *Graphium* spp. and *Colletotrichum gloeosporioides*) and oomycetes (*Phytophthora cinnamomi*) affecting avocado crops [7,8,9]. However, the potential of the associated actinobacteria from avocado rhizosphere to control pathogenic bacteria of diverse crops needs to be explored, e.g., *Xanthomonas* spp. The members of actinobacteria belong to species widely distributed in different habitats and ecological niches and are well known to be a prominent source of bioactive natural products [10,11,12]. Particularly, species from the genus *Streptomyces* are natural antagonist against phytopathogenic fungi and bacteria and exert an inhibitory function by means of different biocontrol mechanisms [13,14].

The antagonism activity of actinobacteria has been shown to be effective in controlling diseases in economically important crops, such as the bacterial panicle blight on rice produced by *Burkholderia glumae* [15,16]; bacterial wilt by *Ralstonia solanacearum* [17,18]; potato common scab by *Streptomyces* spp. [19]; and soft rot by *Dickeya zeae* and *Pectobacterium carotovorum* [20,21]. The secondary metabolites obtained from *Streptomyces* spp. also represent another option for the integral management of diseases caused by phytopathogenic bacteria, but should be used correctly to prevent resistance [22]. Kulkarni et al. [23] demonstrated the antimicrobial activity (AA) against phytopathogenic bacteria and fungi of D-actinomycin produced by *S*. *hydrogenans* IB310. This compound could be useful in plant disease treatment [17,23]. In other studies, cyclic antimicrobial peptide production (dipeptide and tetrapeptide) by *Streptomyces* spp. represents another alternative to control plant pathogenic bacteria [24,25]. Recently, Nguyen et al. [26] demonstrated the potential of *Streptomyces* sp. JCK-8055 and its antibacterial metabolites in the in vitro and in vivo biocontrol of bacterial wilt and fire blight of apple caused by *Erwinia amylovora*. Similarly, Padilla-Gálvez et al. [27] reported that *Streptomyces* sp. TP199 inhibited the in vitro growth of *P*. *carotovorum* subsp. *carotovorum* and *P*. *atrosepticum* in tuber slice maceration trials, which reduced maceration halos caused by these phytopathogenic bacteria.

On the other hand, pretreatment of soil samples with chemical, physical, or combination agents are methods used for selective isolation of actinobacteria [28,29]. Microwave irradiation has been reported as a useful pretreatment in the isolation of actinobacteria by increasing cultured actinobacteria, highlighting that some isolates of the genera *Streptomyces*, *Nocardia*, *Pseudonocardia*, *Amycolatopsis* and *Saccharotrix* were only cultured after microwave irradiation [28,30].

Therefore, to find antagonist actinobacteria that may be useful in the integral management of the bacterial spot disease in chili pepper, the objective of this study is to isolate, identify avocado rhizosphere actinobacteria, and evaluate their anti-*Xanthomonas* activity.

## 2. Materials and Methods

### 2.1. Sampling of Rhizospheric Soil

Rhizospheric soil from an avocado (*Persea americana* Mill.) cv. Hass orchard was collected at the municipality of Ziracuaretiro, Michoacán, México during the dry season in April 2017, as described by Trinidad-Cruz et al. [31]. Two soil samples/tree were taken with a shovel from the rhizosperic area of four trees and deposited into a separated sterile bag. In the laboratory, the subsamples were air-dried at room temperature for one week and then sieved by a 1 mm mesh. Before the bacteria isolation procedure, soil subsamples (5 g each) were vigorously mixed inside a sterile bag for 1 min to obtain a composite sample.

### 2.2. Physical Treatments of Composite Soil Sample

The composite soil sample was separated in two portions or subsamples. One composite subsample (10 g placed in a sterile 15 mL conical tube) was treated for 1 h with dry heat at 70 °C in a drying oven [32]. This sample was left to cool at room temperature for 15 min and deposited in a 160 mL milk dilution bottle with 90 mL of sterile distilled water. The second composite subsample (10 g) was deposited in a sterile 15 mL conical tube with 4 mL of sterile distilled water and sealed. The sealed conical tube was placed into a 1 L beaker with 900 mL of tap water at 4 °C and treated with microwave irradiation according to Wang et al. [30] with slight modifications. The beaker was placed in the center of the microwave oven (2450 MHz, Panasonic^®^ Inverter System Inside, Osaka, Japan) and irradiated for 3 min (100% Power, 1100 W). Thereafter, the sample was transferred to a 160 mL milk dilution bottle with 86 mL of sterile distilled water.

### 2.3. Isolation of Actinobacteria

Each soil suspension (10^−1^ dilution) contained in the milk dilution bottles was stirred manually for 5 min and then 1 mL was transferred to a conical tube with 9 mL of sterile water (10^−2^ dilution) using a micropipette (100–1000 μL). The conical tubes were vortexed for 1 min before continuing with the decimal serial dilutions until the 10^−4^ dilution. The conical tubes of each dilution were mixed by inversion (five to ten times) and then a 100 μL aliquot was spread with a Drigalsky spatula over different culture media with agar in triplicate: yeast-malt extract agar (International *Streptomyces* Project, ISP) medium 2: ISP2: 4 g L^−1^ of yeast extract, 10 g L^−1^ malt extract, 4 g L^−1^ of dextrose, and 20 g L^−1^ of agar, pH 7.3), and oat agar (ISP medium 3: ISP3: 20 g L^−1^ of oats, 1 mL L^−1^ of trace solution [1 g L^−1^ of FeSO_4_·7H_2_O, MnCl_2_·4H_2_O and ZnSO_4_·7H_2_O)] and 18 g L^−1^ of agar, pH 7.3) [33]. All media were supplemented with nalidixic acid (12.5 mg L^−1^) and cycloheximide (50 mg L^−1^) [34]. Petri plates were maintained at 28 °C for 14 d. The amount of cultured actinobacteria was quantified from the plate count of dilutions 10^−3^ or 10^−4^ and expressed as CFU g^−1^ of dry soil. Petri dishes were checked daily from the first week of incubation, focusing on purification of actinobacteria from dilutions 10^−1^ and 10^−2^ and then continued with the remaining dilutions. The emerging actinobacteria-like colonies were subcultured on ISP2 or ISP3 media until pure cultures were obtained. Isolates were preserved as spore suspensions and/or mycelium in glycerol at 25% (*v*/*v*) at −80 °C [35]. The flow diagram of the general process of the origin of the rhizospheric soil samples, pretreatments, isolation and preservation of the actinobacterium-type isolates is illustrated in Appendix A.

### 2.4. Selection of Isolates with Antibacterial Activity (AA)

*Xanthomonas* sp. BV801 was reactivated in nutritional glycerol yeast extract agar (NYGA; 5 g L^−1^ of bactopeptone, 3 g L^−1^ of yeast extract, 20 g L^−1^ of glycerol and 15 g L^−1^ of agar) [36,37] at 28 °C for 2 days. The bacterial suspension used to test antibacterial activity was prepared from a 20 mL bacteria culture grown on NYG broth (NYGB) at 200 rpm/28 °C for 18 to 24 h. Then, strain BV801 was diluted at 1 OD_600nm_ with fresh NYGB for inhibitory assay, as described by Trinidad-Cruz et al. [31].

The AA assay was performed by the double-layer agar method according to Salamoni et al. [38], with slight modifications according to Trinidad-Cruz et al. [31]. All isolates were cultured on ISP2 or ISP3 agar at 28 °C for one week. An agar disk 7 mm in diameter was deposited by the punctual inoculation method [39] on Petri plates with potato dextrose agar (PDA); each Petri plate, containing four different isolates, was maintained at 28 °C for 5 d. Overlay agar was prepared by mixing 400 µL of the pathogenic bacterial suspension (OD_600nm_ = 1) with 4 mL of soft NYGA (6 g L^−1^ of agar) melted and tempered at 48 °C, and then poured over the inoculated PDA Petri plates and maintained at 28 °C for 2 d. Inhibition halos (diameter, inhibition zones = IZ) were measured, considering the center of the agar disc as the central point, towards the margins of *Xanthomonas* sp. BV801 inhibition. When the inhibition halos were mixed, it was measured from the center of the bite towards the inhibition margin of *Xanthomonas* sp. BV801, which was considered as part of the inhibition halo of the isolate. Inhibition halos were measured using a digital vernier. The experiment was performed in triplicate using a completely randomized experiment design.

### 2.5. Effect of Culture Media on Antibacterial Activity of Selected Isolates

Selected isolates, four from the series BVEZ (32, 50, 71, and 73) and four from the series BVEZMW (12, 30, 60, and 81) were selected to evaluate the effect of culture media on the expression of antibacterial activity. The AA assay was performed by the double-layer agar method as described previously. The selected isolates were inoculated individually centered on Petri plates with PDA, ISP2, and ISP3 culture media and incubated at 28 °C for 5 d. The overlay agar suspension with the pathogenic bacteria was prepared and poured over the agar plates as described previously. The Petri plates were maintained at 28 °C for 2 to 3 d. The diameter of the inhibitory halo zone was determined with a digital caliper. All the isolates were evaluated in quadruplicate using a completely randomized distribution.

### 2.6. Molecular Identification and Phylogenetic Analysis of Selected Actinobacteria

A 125 mL Erlenmeyer flask with 20 mL of ISP2 culture media was inoculated with a spore suspension and maintained in agitation at 250 rpm and 28 °C for 3 d. Mycelia cell biomass was twice washed with sterile distilled water, frozen with liquid nitrogen, lyophilized and pulverized at 25 Hz for 5 min (Mixer Mill MM 400, Retsch^®^, Haan, Germany). The genomic DNA was extracted from 1 mg of the pulverized biomass with the Dynabeads^®^ DNA DIRECT^™^ Universal kit (ThermoFisher Scientific, Waltham, MA, USA) following the manufacturer’s instructions. Genomic DNA integrity was confirmed via 1.2% agarose gel electrophoresis in Tris-acetate-EDTA (TAE) buffer and visualized by ultraviolet fluorescence after staining with GelRed^®^ (Biotium, Inc., Fremont, CA, USA). The genomic DNA was stored at −20 °C until use.

The 16S rRNA gene amplification was performed by the polymerase chain reaction (PCR) using the fD1 oligonucleotides (5′-CCGAATTCGTCGACAACAGAGTTTGATCCTGGCTCAG-3′) and rD1 (5′-CCCGGGATCCAAGCTTAAGGAGGTGATCCAGCC-3′) [40]. The PCR was performed in a reaction mixture of 50 µL that contained 5 µL of 10× High Fidelity PCR Buffer; 2 µL of 50 mM of MgSO_4_; 4 µL of 2 mM of dNTPs; 2 µL of each fD1/rD1 oligonucleotide (6 pmol µL^−1^); 0.2 µL of Taq DNA polymerase (5 U µL^−1^; Platinum^®^ Taq DNA High Fidelity; Thermo Fisher Scientific, Waltham, MA, USA); 5 µL of genomic DNA and 29.8 µL of distilled water (UltraPure^™^ DNase/RNase-Free Distilled Water, Thermo Fisher Scientific, Waltham, MA, USA). The PCR was performed in a thermocycler with the following conditions: an initial denaturalization at 94 °C/2 min, followed by 35 cycles at 94 °C/15 s, 58 °C/30 s, and 68 °C/1.5 min, followed by a final extension at 68 °C for 3 min. The PCR conditions for time and temperature were implemented according to the manufacturer’s instructions for the Taq DNA polymerase, except for the alignment temperature. The alignment temperature of the fD1/rD1 primers of 58 °C was obtained from a temperature gradient PCR. The amplification products were confirmed by 1.2% agarose gel electrophoresis on TAE buffer and visualized by ultraviolet fluorescence after GelRed^®^ staining. The amplified fragments were purified starting from agarose gel at 0.8% in TAE buffer with the kit Wizard^®^ SV Gel and PCR Clean-Up System (Promega, Madison, WI, USA). PCR products were sequenced at Macrogen Inc. (Seoul, Republic of Korea, https://dna.macrogen.com/) using the oligonucleotides 800R (5′-TACCAGGGTATCTAATCC-3′), 1100R (5′-GGGTTGCGCTCGTTG-3′) and rD1.

To obtain the consensus of the 16S rRNA gene sequences, the electropherograms of each actinobacteria were assembled, revised and edited with the SnapGene^®^ software (version 4.3.7, GSL Biotech, Chicago, IL, USA). The 16S rRNA gene sequences were compared with other type strain sequences available in the database EzBioCloud 16S (https://www.ezbiocloud.net/) [41]. Subsequently, the closest type strain sequences and similarity values of the 16S rRNA gene sequences (calculated based on pair sequence alignment) of the 16S-based ID application were identified in EzBioCloud. The sequences of the type strains were selected based on the percentage of similarity of the 16S rRNA gene greater than 99%. Multiple alignments of 16S rRNA gene sequences of the actinobacteria and closest sequences of type strains were performed with MUSCLE. The phylogenetic tree was constructed by the neighbor-joining (NJ) method based on the nucleotide substitution model of Kimura-two parameter using the MEGA11 software [42,43,44]. The topology of the phylogenetic tree was evaluated by the bootstrap analysis using 1000 resamples [45]. The 16S rRNA gene sequence of *Rubrobacter radiotolerans* DSM 5868^T^ was used as the outgroup.

The 16S rRNA gene sequences of the actinobacteria BVEZ 32 (OP677868), BVEZ 50 (OP677869), BVEZ 71 (OP677870), BVEZ 73 (OP677871), BVEZMW 12 (OP677872), BVEZMW 30 (OP677873), BVEZMW 60 (OP677874), and BVEZMW 81 (OP677875) were deposited in the GenBank database.

### 2.7. Statistical Analyses

The normal distribution and homoscedasticity of residual data were evaluated by Shapiro–Wilk and Bartlett tests, respectively. An analysis of variance (ANOVA) and Tukey’s multiple comparison test (*p* ≤ 0.05) were used to analyze the data. Student’s *t*-tests of independent samples were used to compare mean bacterial concentration (CFU g^−1^ of dry soil) between pretreatments and culture media. All the statistical procedures were performed using StatGraphics Centurion XVI software (version 16.2.04, Statgraphics Technologies, Inc., The Plains, VA, USA). The data of the inhibition halos on ISP2 agar were transformed using the following equation:√(x + 0.5)

## 3. Results

### 3.1. Soil Treatments to Improve Isolation of Actinobacteria

Initially, soil rhizospheric samples were subjected to microwave irradiation and dry heat treatments and serial dilutions were seeded onto ISP2 and ISP3. Thereafter, the number of actinobacteria-like isolates were clearly increased when soil samples were subjected to microwave irradiation (Table 1). The results also showed a higher number of isolates when ISP3 agar media were used.

Microwave irradiation increased the concentration of cultured actinobacteria on ISP2 (2.1-fold increase) and ISP3 (3.6-fold increase) agar compared to samples pretreated with dry heat (Figure 1). The counts of culturable actinobacteria were similar for ISP2 and ISP3 agar when the soil sample was pretreated with dry heat, but when irradiated with microwaves, the count increased in ISP3 agar (1.6-fold increase) with respect to ISP2 agar (Figure 1).

### 3.2. Antibacterial Activity of the BVEZ and BVEZMW Isolates

The results of antibacterial activity (AA) over the pathogen *Xanthomonas* sp. BV801 showed that 19/76 (25%) isolates (BVEZ) and 24/122 (19.7%) (BVEZMW) strains isolated from samples treated by dry heat and microwave irradiation were obtained, respectively (Table 2). It is shown that microwave irradiation treatment enhanced the isolation of actinobacteria in both culture media. The AA between BVEZ actinobacteria showed significant differences (Tukey’s test, *p* ≤ 0.05); three isolates (BVEZ 32, BVEZ 50, and BVEZ 71) showed the highest IZ, with values up to 30 mm in diameter (Figure 2a). Also, it was noticeable that 18 isolates, including the strains mentioned above, were isolated using ISP3 agar media. The increment in the number of isolates by microwave treatment does not correlate with finding isolates with superior AA. The isolate BVEZMW 12 showed the greatest IZ (55.47 ± 1.57 mm in diameter); significant differences (Tukey’s, *p* ≤ 0.05) were found in reference to the other isolates from the same series (Figure 2b). The above evidence showed that actinobacteria isolates from soil treated by microwave irradiation showed a lower AA than isolates from soil treated with dry heat; in addition, different isolates (species) were found with both isolation methods.

### 3.3. Effect of Culture Media for Secondary Metabolite Production

The effect of culture media composition on the production of inhibitory metabolites on selected isolates were evaluated. Four isolates from the dry heat treatment (BVEZ 32, BVEZ 50, BVEZ 71, and BVEZ 73) and four strains isolated from soil samples treated with microwave irradiation (BVEZMW 12, BVEZMW 30, BVEZMW 60, and BVEZMW 81) were seeded on PDA, ISP2, and ISP3 agar media, and inhibitory activity over *Xanthomonas* sp. BV801 was evaluated (Figure 3). In general, PDA culture media induced the production of secondary metabolites with inhibitory activity against *Xanthomonas* sp. in all the isolates; ISP3 agar media induced the production of bioactive metabolites of the isolates with lesser production than isolates grown on PDA media (Figure 3 and Figure 4). Expression of inhibitory metabolites was repressed on isolates BVEZ 32, BVEZ 71 and BVEZ 73 when grown on ISP2 media. Furthermore, we observed at least a three-fold increment in the inhibitory activity of the selected isolates from the series BVEZ when grown on PDA media. Statistical differences in inhibitory activity of the BVEZMW isolates, which grew on the three-culture media, were observed.

In brief, PDA was the media when the inhibitory metabolites of all isolates were induced; on the contrary, the isolates inoculated on ISP2 and ISP3 culture media suffered from a certain repressive effect on their bioactive metabolite production (Figure 4). In addition, it is clear that there was an enhanced effect of antibacterial activity of isolates in the BVEZ series rather than the BVEZMW series.

### 3.4. Identification of the Selected Actinobacteria

The analysis of the 16S rRNA gene sequences of the isolates BVEZMW 30 (1331 bp), BVEZ 32 (1428 bp), BVEZ 50 (1497 bp), BVEZ 71 (1516 bp), BVEZ 73 (1277 bp), and BVEZMW 81 (1399 bp) revealed a close relationship with species of genus *Streptomyces*. Meanwhile, isolates BVEZMW 12 and BVEZMW 60 showed a high sequence similarity with the 16S rRNA gene sequence of genus *Amycolatopsis*.

The phylogenetic analysis based on the 16S rRNA gene sequence showed two identified clades; one included the *Amycolatopsis* species (BVEZMW12 and BVEZMW60), and one clade encompassed the *Streptomyces* species. In the *Streptomyces*-clade, there are four separate identified subclades. One group includes the isolates BVEZ 32, BVEZ71, and BVEZ73; groups 2 to 4 include the isolates BVEZ50, BVEZMW30, and BVEZMW81, respectively (Figure 5).

The BVEZMW 12 and BVEZMW 60 isolates showed the closest relationship with the strain type *A*. *speibonae* JS72^T^. Meanwhile, BVEZMW 30 was closely related with *S*. *neopeptinius* KNF 2047^T^ and *S*. *cyslabdanicus* K04-0144^T^. The BVEZ 50 isolate shared a high sequence similarity with *S*. *olivicoloratus* T13^T^ and *S*. *fuscichromogenes* m16^T^. In the case of the BVEZ 71 isolate, it was grouped with *S*. *graminisoli* JR-19^T^, *S*. *hyaluromycini* MB-PO13^T^, and *S*. *shenzhenensis* subsp. *shenzhenensis* 172115^T^. Finally, the isolate BVEZMW 81 was related with *S*. *brevispora* BK160^T^ and *S*. *laculatispora* BK166^T^.

## 4. Discussion

The results indicated that microwave irradiation (2450 MHz, 1100 W for 3 min) increased the amount of cultured actinobacteria (CFU g^−1^ from dry soil) and the number of isolates on ISP2 and ISP3 agar compared to dry heat pretreatment. Wang et al. [30] found that microwave irradiation with 120 W power per 3 min to soil samples increased the amount of culturable actinobacteria in the three media evaluated compared to non-irradiated samples. Similarly, Hamedi et al. [46] reported that the highest percentage of isolates was obtained from microwave-irradiated soil samples, followed by dry heat pretreatment at 120 °C. However, our conditions differ from previous studies in the power used during microwave irradiation, as well as in the temperature of the water prior to irradiation of the soil sample. In our study, we found that some actinobacteria of the genera *Streptomyces* and *Amycolatopsis* were only culturable after microwave irradiation. Similarly, Wang et al. [30] reported that some isolated *Streptomyces* were only cultured after microwave irradiation of soil samples. In controlled experiments, Komarova et al. [47] indicated that microwave irradiation (80 W power per 1 min) of soil added with *S*. *xanthochromogenes* increased spore germination with respect to the non-irradiated sample. In another study, Likhacheva et al. [48] found that under controlled conditions, microwave irradiation (2450 MHz, 80 W power, 15 to 90 s) suppressed or stimulated the growth and development of *Streptomyces* spp. In addition, the authors indicated that microwave irradiation as a pretreatment can change the behavior of actinobacteria in the microbiome of soil samples, similar to the results of this study and what was reported by Wang et al. [30].

Searching for antagonist microorganisms is an important strategy that, along with others, allows the integral management of diseases caused by phytopathogen species of the genus *Xanthomonas* [3,6]. This study investigated the antibacterial activity (AA) of actinobacteria isolated from avocado rhizosphere against *Xanthomonas* sp. BV801, which causes chili pepper bacterial spots. Among the isolates, four BVEZ (32, 50, 71 and 73) actinobacteria showed the greatest AA by the method of punctual inoculation in vitro inhibiting growth of *Xanthomonas* sp. BV801 almost completely with inhibition zones (IZ) that fluctuated from 62.2 to 80.2 mm in diameter in PDA medium. Similar results have been determined by Sharma and Thakur [49], who isolated 77 actinobacteria of the forest ecosystem soil, among which PB-69 and Kz-32 recorded the greatest AA with IZ of 60 and 70 mm in diameter by the punctual inoculation method against *Staphylococcus aureus* MTCC 96, respectively. Similarly, in our previous study, the actinobacteria isolated from the rhizosphere of avocado cv. Hass trees showed antimicrobial activity against *C*. *gloeosporioides* and *Xanthomonas* sp. BV801 without pretreatment on soil [31]. Among these actinobacteria, 12 of 41 isolates showed AA with IZ from 15.5 to 62.7 mm in diameter [31].

In a recent study, Vega-Torres et al. [9] reported the antagonistic activity of 117 actinobacteria isolated from the rhizosphere of avocado cv. Hass trees in the municipality of Xalisco, Nayarit against *F*. *oxysporum*, among which the A31 isolate stood out because of its antifungal activity and was identified based on the 16S rRNA gene sequence as *S*. *olivicoloratus* [9]. Likewise, in this study, the BVEZ 50 actinobacteria that showed a strong AA demonstrated a high sequence similarity to *S*. *olivicoloratus* T13^T^ (99.79%). In a previous study, Nguyen and Kim [50] reported the antimicrobial activity of *S*. *olivicoloratus* T13^T^, isolated from forestall soil in South Korea against Gram-positive bacteria (*B*. *subtilis* KEMB 51201-001, *St*. *aureus* KEMB 4659, *St*. *epidermidis* KACC 13234, and *Paenibacillus larvae* KACC 14031), Gram-negative bacteria (*P*. *aeruginosa* KACC 10185, and *Escherichia coli* KEMB 212-234), and fungi (*Candida albicans* KACC 30003 and *Aspergillus niger* KACC 40280). The wide range of antimicrobial activity of *S*. *olivicoloratus* T13^T^ agrees with the results of this study with respect to the AA of *Streptomyces* sp. BVEZ 50 against *Xanthomonas* sp. BV801 and the antifungal activity of *Streptomyces* sp. A31 against *F*. *oxysporum*, according to that reported by Vega-Torres et al. [9].

*Streptomyces*—generally the most abundant genus of soil—is a source of metabolites with AA [49,51,52]. The phylogenetic analysis based on the 16S rRNA gene sequences of the BVEZ (32, 50, 71, and 73), BVEZMW 30 and BVEZMW 81 actinobacteria sequences indicated they belong to the genus *Streptomyces*. *Streptomyces* sp. BVEZ 32 and *Streptomyces* sp. BVEZ 73, which showed outstanding antibacterial activity, were grouped in one clade with type strain of *S*. *graminisoli* JR-19^T^ in the NJ phylogenetic tree. Interestingly, the results of this study agree with that reported by Han et al. [53], where 12 of 50 *Streptomyces* spp. isolated from the bamboo (*Sasa borealis*) rhizospheric soil showed strong AA against *X*. *axonopodis* pv. *vesicatoria* KACC 12872, which causes chili pepper bacterial spots. Among these *Streptomyces*, one of them was identified as a new species of the genus *Streptomyces* proposed as *S*. *graminisoli* JR-19^T^ [54]. Additionally, other studies have reported the AA of *S*. *graminisoli* strains against non-phytopathogenic bacteria, such as *S*. *graminisoli* AC12 against *Rhizobium tropici* [55] and *S*. *graminisoli* PWS11 against *St*. *aureus* MTCC 96 and *E*. *coli* MTCC 40 [51]. Nevertheless, the answer to what secondary metabolites of *S*. *graminisoli* are related with their antibacterial activity is still unknown.

In the NJ phylogenetic tree, *Streptomyces* sp. BVEZ 71 was more closely related to *S*. *shenzhenensis* subsp. *shenzhenensis* 172115^T^ and *S*. *shenzhenensis* subsp. *oryzicola* W18L9^T^. *S*. *shenzhenensis* subsp. *shenzhenensis* 172115^T^, isolated from the mangrove soil, have not been reported with AA [56,57], while for *S*. *shenzhenensis* subsp. *oryzicola* W18L9^T^, Kaewkla et al. [58] recently recorded their AA against *X*. *oryzae* pv. *oryzae* PXO 86 in the HPDA (half-strength PDA) culture medium but not in ISP2. Similar results were found in this study with *Streptomyces* sp. BVEZ 71, which showed AA in the PDA medium (IZ of 75.8 mm in diameter) but not in ISP2. Subsequent studies with a polyphasic focus are necessary to determine the taxonomic position of *Streptomyces* sp. BVEZ 71, as well as the secondary metabolites implied in their AA.

On the other hand, *Streptomyces* sp. BVEZMW 81 was grouped based on the phylogenetic analysis in a clade with *S*. *brevispora* BK160^T^. The original description of *S*. *brevispora* BK160^T^ was from a hay lot soil isolate, and its possible antimicrobial potential is not indicated [59]. Nevertheless, Zhao et al. [60] isolated 88 endophyte actinobacteria from plants of *Glycyrrhiza inflata*; among these isolates, *Streptomyces* sp. SCAU5210 showed AA against *E*. *coli* ATCC 35218 (15.5 mm) but not against *St*. *aureus* ATCC 25923 and a weak activity against phytopathogen fungi. *Streptomyces* sp. SCAU5210 showed a close relationship with the type strain of *S*. *brevispora* BK160^T^ based on the analysis of sequence similarity and phylogenetics of the 16S rRNA gene sequence. In this study, *Streptomyces* sp. BVEZMW 81 showed the greatest AA (44.88 ± 1.04 mm in diameter) against *Xanthomonas* sp. BV801 in ISP2 medium.

On the other hand, *Streptomyces* sp. BVEZMW 30 was grouped in a clade with *S*. *neopeptinius* KNF 2047^T^. In previous studies, Kim et al. [61] demonstrated the production of antifungal compounds identified as neopeptines A and B by *S*. *neopeptinius* KNF 2047^T^. These compounds inhibited spore germination of *Cladosporium cucumerinum*, *Colletotrichum lagenarium* (=*C*. *orbiculare*), and *Magnaporthe grisea* with a minimum concentration of inhibition values from 128 to 256 µg mL^−1^, while phytopathogen bacteria, such as *Erwinia carotovora* pv. *carotovora* and *R*. *solanacearum,* did not show activity, including at 512 µg mL^−1^. Similarly, Han et al. [62] reported that the use of bacterial culture of *S*. *neopeptinius* KNF 2047^T^ (biomass and broth) did not inhibit growth of *B*. *subtilis* and *Salmonella typhimurium*. The results in this research on the AA of *Streptomyces* sp. BVEZMW 30 differ with that reported for *S*. *neopeptinius* KNF 2047^T^. The other type strains of *Streptomyces* that showed sequence similarity with *Streptomyces* sp. BVEMW 30 as *S*. *cyslabdanicus* K04-0144^T^ (99.1%), *S*. *olivaceoviridis* NBRC 13066^T^ (99.1%), and *S*. *canarius* NBRC 13431^T^ (99.02%) were grouped in a different clade. Based on the results, *Streptomyces* sp. BVEZMW 30 could represent a new taxon of the genus *Streptomyces*. Nevertheless, further studies are required with a polyphasic focus to determine the taxonomic position of *Streptomyces* sp. BVEZMW 30.

The genus *Amycolatopsis*, which belongs to the family *Pseudonocardiaceae,* contains several species of relevance because of the production of bioactive metabolites of pharmaceutical importance according to the review by Tan and Goodfellow [63]. In this study, the phylogenetic relationships with members of the genus *Amycolatopsis* and the AA against *Xanthomonas* sp. BV801 were grouped in one clade with *A*. *speibonae* JS72^T^. In 2014, Everest et al. [64] reported the AA of *A*. *speibonae* JS72^T^ against *Enterococcus faecalis* (strain sensitive to vancomycin), *E*. *phoeniculicola* JLB-1^T^, *Mycobacterium aurum* A+ and *St*. *aureus* ATCC 25923, while for *E*. *coli* ATCC 25922 and *E*. *faecium* VanA (strain resistant to vancomycin), they did not show activity. Moreover, the presence of antibiotic biosynthetic genes related to antibiotic production, such as glycopeptides and ansamycins, has been detected in *A*. *speibonae* JS72^T^ [64].

In this study, actinobacteria from avocado tree rhizosphere demonstrated to be a potential agent against *Xanthomonas* sp. BV801. Among these actinobacteria isolated, *Streptomyces* spp. and *Amycolatopsis* spp. showed strong AA against *Xanthomonas* sp. BV801 in in vitro conditions. Further studies are necessary to specifically detect what secondary metabolites of those isolated from *Streptomyces* spp. and *Amycolatopsis* spp. imply in terms of *Xanthomonas* sp. BV801 growth inhibition. Some *Streptomyces,* such as *Streptomyces* sp. BVEZMW 30 and *Streptomyces* sp. BVEZMW 81, are potentially new biological control agents of *Xanthomonas*, due to the unknown antibacterial activity against phytopathogenic bacteria of the closest species shown in the phylogenetic tree (Figure 5). Some actinobacteria (*Streptomyces* spp. and *Amycolatopsis* spp.) were only culturable after microwave irradiation of the soil sample. To understand how *Streptomyces* and *Amycolatopsis* species have the potential to be biocontrol agents beyond their in vitro antibacterial activity, future studies should focus on issues of interactions of chili pepper plants with the inoculation of these actinobacteria or the application of their metabolites when they are infected with *Xanthomonas*. Other studies are necessary to evaluate their potential use in agriculture, such as determining the concentration, formulation and type of application (foliar or soil) of these actinobacteria (spores or bioactive metabolites) for in vivo control of this foliar disease caused by *Xanthomonas* spp., with experiments in greenhouse conditions and their validation in field trials.

## Figures and Tables

**Figure 1 microorganisms-12-02199-f001:**
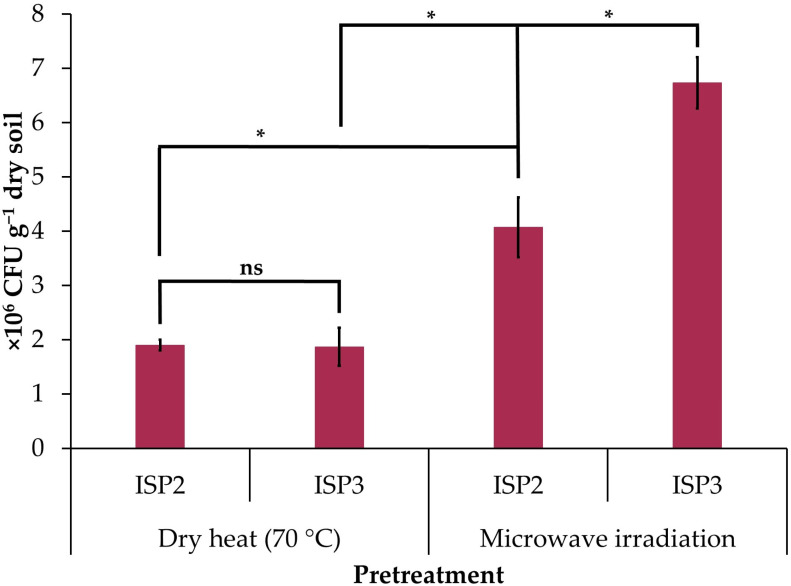
Concentration of cultured actinobacteria in the soil sample of the avocado rhizosphere pretreated with dry heat (70 °C for 1 h) and microwave irradiation (1100 W for 3 min) in the ISP2 and ISP3 agar culture media. The data are the mean ± standard deviation. Asterisks (*) indicate significant differences between culture media or pretreatments according to Student’s *t*-test (*p* ≤ 0.05). NS: not significant.

**Figure 2 microorganisms-12-02199-f002:**
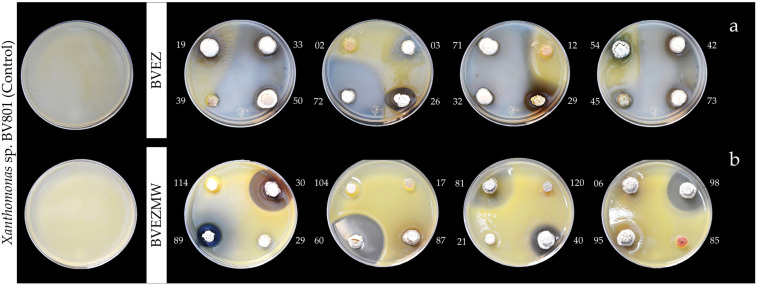
Image of initial screening of in vitro antibacterial activity of actinobacteria performed by the punctual inoculation and double-layer agar: (**a**) BVEZ isolates; (**b**) BVEZMW isolates.

**Figure 3 microorganisms-12-02199-f003:**
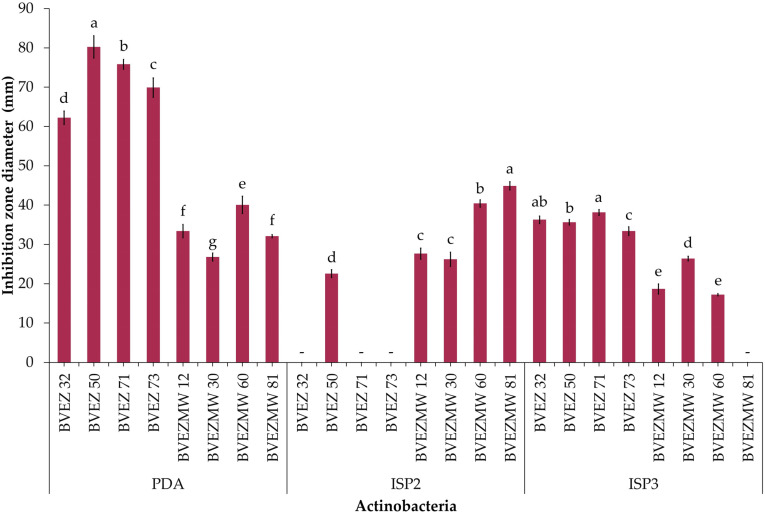
Effect of culture medium (PDA, ISP2 and ISP3) on the antibacterial activity of selected actinobacteria of the BVEZ and BVEZMW series against *Xanthomonas* sp. BV801. The diameter data of the inhibition zone represent the average of four biological repetitions. -: not detected. Different letters on each bar for each culture medium indicate significant differences among the media according to Tukey’s test (*p* ≤ 0.05).

**Figure 4 microorganisms-12-02199-f004:**
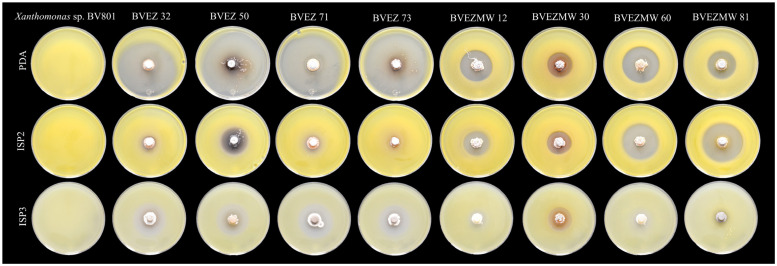
Representative image of the effect of culture medium (PDA, ISP2 and ISP3) on the antibacterial activity of eight selected actinobacteria from the BVEZ and BVEZMW series against *Xanthomonas* sp. BV801 by the double-layer method of agar.

**Figure 5 microorganisms-12-02199-f005:**
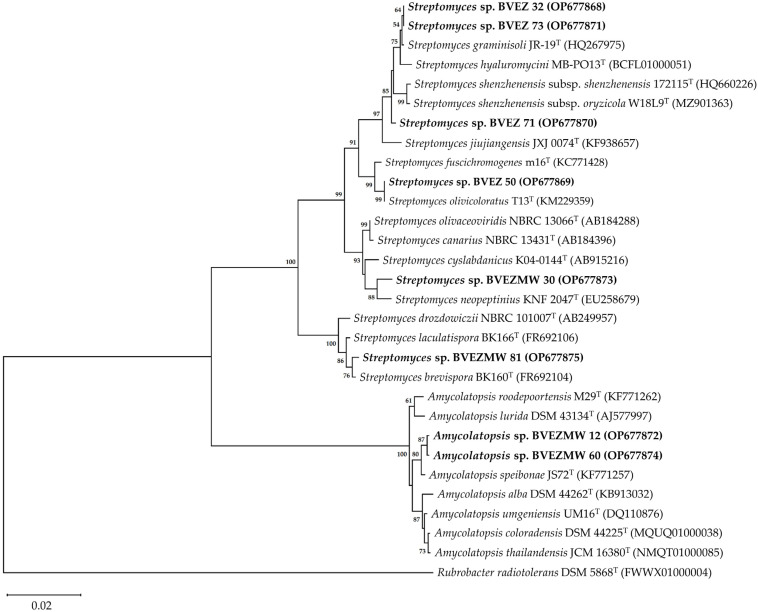
Phylogenetic neighbor-joining tree based on the 16S rRNA gene sequence showing the phylogenetic relationships among the BVEZ (32, 50, 71, and 73), BVEZMW (12, 30, 60, and 81) actinobacteria and type strains of the genera *Streptomyces* and *Amycolatopsis*. The bootstrap values (>50%) are shown next to the branches. The sequence of *Rubrobacter radiotolerans* DSM 5868^T^ was used as an external group. The access numbers to the GenBank sequences are shown in parentheses. Bar = 0.02 substitutions by nucleotide position.

**Table 1 microorganisms-12-02199-t001:** Effect of physical pretreatment to soil samples on number of actinobacteria isolated.

Soil Pretreatment	Culture Medium	Number of Isolates	Code
Dry heat (70 °C)	ISP2	23	BVEZ 01–23
	ISP3	53	BVEZ 24–76
Microwave irradiation	ISP2	61	BVEZMW 01–61
	ISP3	61	BVEZMW 62–122

**Table 2 microorganisms-12-02199-t002:** Zones of inhibition (inhibition halos) of *Xanthomonas* sp. BV801 obtained during the antibacterial activity test in PDA by the actinobacteria of the BVEZ and BVEZMW series.

Dry Heat		Microwave Irradiation	
Isolate	Inhibition Zone (mm) ^†^	Isolate	Inhibition Zone (mm) ^†^
BVEZ 03	12.76 ± 1.02 ^k^*	BVEZMW 03	27.65 ± 2.16 ^d^*
BVEZ 26	20.34 ± 0.54 ^hi^	BVEZMW 05	20.98 ± 1.33 ^efg^
BVEZ 28	54.08 ± 2.06 ^cd^	BVEZMW 08	28.48 ± 0.49 ^d^
BVEZ 29	38.82 ± 0.17 ^f^	BVEZMW 12	55.47 ± 1.57 ^a^
BVEZ 31	23.79 ± 0.67 ^h^	BVEZMW 19	12.22 ± 0.68 ^jk^
BVEZ 32	82.37 ± 0.78 ^a^	BVEZMW 30	34.63 ± 0.39 ^c^
BVEZ 33	48.23 ± 2.11 ^e^	BVEZMW 31	21.94 ± 1.54 ^ef^
BVEZ 40	16.67 ± 1.05 ^ijk^	BVEZMW 37	15.71 ± 0.36 ^ij^
BVEZ 42	55.15 ± 2.38 ^cd^	BVEZMW 40	23.12 ± 1.44 ^e^
BVEZ 46	19.69 ± 1.51 ^hij^	BVEZMW 50	16.36 ± 0.89 ^hi^
BVEZ 50	70.64 ± 2.05 ^b^	BVEZMW 51	11.40 ± 0.35 ^k^
BVEZ 64	53.21 ± 1.36 ^d^	BVEZMW 60	44.57 ± 0.58 ^b^
BVEZ 65	22.35 ± 1.47 ^h^	BVEZMW 68	21.80 ± 1.28 ^ef^
BVEZ 67	48.20 ± 0.95 ^e^	BVEZMW 70	28.75 ± 0.61 ^d^
BVEZ 68	33.53 ± 2.15 ^g^	BVEZMW 74	21.71 ± 1.49 ^ef^
BVEZ 71	66.13 ± 1.78 ^b^	BVEZMW 75	19.35 ± 1.24 ^fgh^
BVEZ 72	52.19 ± 1.88 ^de^	BVEZMW 76	17.69 ± 1.35 ^ghi^
BVEZ 73	58.10 ± 1.57 ^c^	BVEZMW 81	37.24 ± 0.85 ^c^
BVEZ 76	15.60 ± 0.77 ^jk^	BVEZMW 88	16.83 ± 1.37 ^hi^
		BVEZMW 89	18.94 ± 1.40 ^fghi^
		BVEZMW 98	33.96 ± 0.90 ^c^
		BVEZMW 99	37.13 ± 0.47 ^c^
		BVEZMW 107	28.41 ± 0.95 ^d^
		BVEZMW 117	17.32 ± 0.73 ^hi^
		BVEZMW 121	17.63 ± 0.99 ^ghi^

^†^ Inhibition zone diameter data represent the mean of three biological repetitions ± standard deviation (SD). * Data followed by different letters in superscript differ significantly according to Tukey’s (*p* ≤ 0.05) test.

## Data Availability

rRNA 16S gene sequences of BVEZ 32, BVEZ 50, BVEZ 71, BVEZ 73, BVEZMW 12, BVEZMW 30, BVEZMW 60, and BVEZMW 81 were deposited in the GenBank database.

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
