# Peer review of "Isolation, Antibacterial Activity and Molecular Identification of Avocado Rhizosphere Actinobacteria as Potential Biocontrol Agents of Xanthomonas sp."

_microorganisms, 2024, doi:10.3390/microorganisms12112199_

Round 1

Reviewer 1 Report

Comments and Suggestions for Authors

This manuscript discusses the isolation and antibacterial activity of Actinobacteria, specifically Streptomyces and Amycolatopsis, from the avocado rhizosphere. These bacteria are tested as biocontrol agents against Xanthomonas sp., a plant pathogen that affects important crops like chili peppers. The study has potential agricultural relevance, especially for managing bacterial diseases. The research provides useful insights into isolating actinobacteria with antibacterial potential. Since bacterial pathogens like Xanthomonas spp.pose global challenges, finding alternative biocontrol methods is important. The manuscript has a solid foundation, but the approach's novelty and findings must be highlighted. Although similar studies exist, this research adds value, mainly through its use of microwave treatment.

1.      The authors should clarify how this research stands out from previous studies. Since this type of research has been published several times, it’s crucial to show how this work significantly contributes to current scientific knowledge.

2.      The literature review could be expanded to include recent advances in biocontrol methods. The authors should clearly explain how this study contributes to the field. Detailed comparisons with previous studies would help emphasize the importance of the findings.

3.      A more in-depth discussion of the evolutionary relationships between the strains, especially how related species of Streptomyces and Amycolatopsis show antibacterial activity, would improve the discussion. This could also highlight the potential for discovering new species or biocontrol agents within these groups.

4.      The manuscript would benefit from a critical analysis of the study's limitations, such as the in vitro nature of the experiments and the challenges of applying these findings in field conditions. Additionally, discussing environmental factors that might affect the effectiveness of these biocontrol agents in agriculture would strengthen the paper.

Author Response

Author Response (Reviewer 1)

Dear Reviewer,

Thank you for your time and kindness in reviewing our manuscript, as well as your valuable suggestions. Below, we provide you with the answer to each of your comments.

Reviewer 1

Comments and Suggestions for Authors

This manuscript discusses the isolation and antibacterial activity of Actinobacteria, specifically Streptomyces and Amycolatopsis, from the avocado rhizosphere. These bacteria are tested as biocontrol agents against Xanthomonas sp., a plant pathogen that affects important crops like chili peppers. The study has potential agricultural relevance, especially for managing bacterial diseases. The research provides useful insights into isolating actinobacteria with antibacterial potential. Since bacterial pathogens like Xanthomonas spp.pose global challenges, finding alternative biocontrol methods is important. The manuscript has a solid foundation, but the approach's novelty and findings must be highlighted. Although similar studies exist, this research adds value, mainly through its use of microwave treatment.

  1. The authors should clarify how this research stands out from previous studies. Since this type of research has been published several times, it’s crucial to show how this work significantly contributes to current scientific knowledge.

Answer. We agree. Some contributions were indicated regarding microwave treatment, such as using a higher power to isolate actinobacteria (1100 W), while in other studies the power is much lower.

Lines 329-341:

The results indicated that microwave irradiation (2450 MHz, 1100 W for 3 min) increased the amount of cultured actinobacteria (CFU g−1 from dry soil) and the number of isolates on ISP2 and ISP3 agar compared to dry heat pretreatment. Wang et al. [30] found that microwave irradiation with 120 W power per 3 min to soil samples increased the amount of culturable actinobacteria in the three media evaluated compared to non-irradiated samples. Similarly, Hamedi et al. [46] reported that the highest percentage of isolates was obtained from microwave-irradiated soil samples, followed by dry heat pretreatment at 120 °C. However, our conditions differ from previous studies in the power used during microwave irradiation, as well as in the temperature of the water prior to irradiation of the soil sample. In our study we found that some actinobacteria of the genera Streptomyces and Amycolatopsis were only culturable after microwave irradiation. Similarly, Wang et al. [30] reported that some isolated Streptomyces were only cultured after microwave irradiation of soil samples.

  1. The literature review could be expanded to include recent advances in biocontrol methods. The authors should clearly explain how this study contributes to the field. Detailed comparisons with previous studies would help emphasize the importance of the findings.

Answer.: We agree. Two recent studies were added, 2024 and 2021, where they show the importance of Streptomyces and its metabolites in the in vitro and in vivo biocontrol of bacterial diseases.

Lines 76-81: Recently, Nguyen et al. [26] demonstrated the potential of Streptomyces sp. JCK-8055 and its antibacterial metabolites in the in vitro and in vivo biocontrol of bacterial wilt and fire blight of apple caused by Erwinia amylovora. Similarly, Padilla-Gálvez et al. [27] reported that Streptomyces sp. TP199 inhibited the in vitro growth of P. carotovorum subsp. carotovorum and P. atrosepticum in tuber slice maceration trials, which reduced maceration halos caused by these phytopathogenic bacteria.

  1. A more in-depth discussion of the evolutionary relationships between the strains, especially how related species of Streptomyces and Amycolatopsis show antibacterial activity, would improve the discussion. This could also highlight the potential for discovering new species or biocontrol agents within these groups.

Answer: We agree. However, in the discussion it was indicated that the type strains related to our isolates showed antibacterial or antifungal activity or have not been reported any antimicrobial activity, even if the metabolite in question was known, our isolates could or could not produce it until the compound is isolated and identified to confirm this possibility or have the sequence of the biosynthetic gene groups to compare them with others already reported. As the reviewer comments, the results highlight a potential to discover new biocontrol agents.

The following was added:

Lines 448-451: Some Streptomyces such as Streptomyces sp. BVEZMW 30 and Streptomyces sp. BVEZMW 81 are potentially new biological control agents of Xanthomonas, due to the unknown antibacterial activity against phytopathogenic bacteria of the closest species shown in the phylogenetic tree (Figure 5).

  1. The manuscript would benefit from a critical analysis of the study's limitations, such as the in vitro nature of the experiments and the challenges of applying these findings in field conditions. Additionally, discussing environmental factors that might affect the effectiveness of these biocontrol agents in agriculture would strengthen the paper.

Answer: We agree. We consider indicating it as a conclusion:

Liness 451-461: Some actinobacteria (Streptomyces spp. and Amycolatopsis spp.) were only culturable after microwave irradiation of the soil sample. To understand how Streptomyces and Amycolatopsis species have the potential to be biocontrol agents beyond their in vitro antibacterial activity, future studies should focus on issues of interaction of chili pepper plants with the inoculation of these actinobacteria or the application of their metabolites when they are infected with Xanthomonas. Other studies are necessary to evaluate their potential use in agriculture, such as determining the concentration, formulation and type of application (foliar or soil) of these actinobacteria (spores or bioactive metabolites) for in vivo control of this foliar disease caused by Xanthomonas spp. with experiments in greenhouse conditions and their validation in field trials.

Reviewer 2 Report

Comments and Suggestions for Authors

Overall, the paper is scientifically sound, but improvements in clarity, particularly in the discussion and graphical representation, would make the work more accessible and impactful.

  • The introduction is well-written and contextualizes the problem. However, it could benefit from additional references to recent studies on biocontrol agents or innovative methods for isolating actinobacteria. The transition between problems in chili pepper crops and the use of actinobacteria could be smoother.
  • Including a visual representation (flowchart) of the experimental design could improve clarity for readers.
  • The results are presented clearly with adequate statistical support. One suggestion is to provide more detailed comparisons or discussions between the microwave-irradiated and dry-heat-treated samples. Consider expanding on why microwave irradiation yielded more isolates but fewer potent ones.
  • The discussion is comprehensive but could benefit from further explanation on how the findings compare with other studies in the field. For example, a more in-depth analysis of how these results impact biocontrol practices in agriculture could be included. It would also help to discuss any practical applications or future research directions based on the findings.
  • The figures are useful, but some figures and tables (such as Figure 3) could benefit from clearer labels or more detailed explanations in the captions. There seems to be an issue with the labeling of figures ("two Figure 3" references). Please ensure proper figure numbering and cross-referencing.

My special attention was on methodology part:

  1. Sampling of Rhizospheric Soil (Section 2.1):
    • Clarification on Soil Samples: While the sampling method is generally clear, the description of soil collection (e.g., “Two-soil samples/tree were taken...”) could benefit from more details. Were these samples taken at a specific depth? Was there any particular season or environmental condition considered? These factors can influence microbial populations.
    • Homogenization of Subsamples: The text mentions mixing subsamples to obtain a composite sample. It would be helpful to describe how the homogenization process was done to ensure even distribution of microbes (e.g., vortexing or mechanical mixing).
  2. Physical Treatments of Composite Soil Sample (Section 2.2):
    • Heat Treatment Protocol: The protocol for the dry heat method is sufficiently described, but the cooling process after heating is vague (“left to cool at room temperature”). It might be more precise to mention the exact cooling duration or the room temperature range to maintain consistency.
    • Microwave Treatment: The microwave irradiation method needs more clarification on why the specific power and duration (100% power for 3 minutes) were chosen. Was this based on a previous study or preliminary experiments? Additionally, it would be beneficial to discuss potential variations (e.g., microwave type or power output) that could influence reproducibility.
    • Sterile Controls: Were sterile controls included for both treatments to ensure no contamination was introduced during the physical treatment processes? Mentioning sterile controls for the soil treatments would strengthen the robustness of the methods.
  3. Isolation of Actinobacteria (Section 2.3):
    • Dilution Process: The serial dilution method is commonly used, but the text could be more specific about how the samples were mixed and plated to avoid biases (e.g., vortexing prior to each dilution or using calibrated pipettes).
    • Choice of Culture Media: The media used (ISP2 and ISP3) are appropriate, but the rationale for choosing these particular media over others could be explained better. Did preliminary tests show that these media were the most effective for actinobacteria isolation? Were other media tested or considered?
    • Incubation Period: A 14-day incubation period is standard for actinobacteria, but adding details about how the colonies were checked during this period (e.g., intervals for visual inspections) could improve the replicability.
  4. Selection of Isolates with Antibacterial Activity (Section 2.4):
    • Standardization of Pathogen Inoculum: The OD600nm of 1.0 for the Xanthomonas suspension is mentioned, but it’s unclear if this represents the optimal concentration for consistent antibacterial activity measurements. Providing information on how this concentration was determined (e.g., preliminary tests or literature standards) would improve methodological transparency.
    • Antibacterial Activity Assay: The double-layer agar method is well-described, but you could clarify how the measurement of the inhibition zones was standardized. Was the zone measured at the widest point or as an average across different angles? Adding this information will ensure that others can reproduce the results accurately.
  5. Effect of Culture Media on Antibacterial Activity of Selected Isolates (Section 2.5):
    • Selection of Culture Media: The rationale behind using PDA, ISP2, and ISP3 media for secondary metabolite production needs to be explained in more depth. Why were these specific media chosen, and how do they compare to other media that might have been used in similar studies?
    • Replicability of Inoculation: While the inoculation procedure is described well, it might help to include details about how inoculum consistency was maintained across all media types (e.g., consistent spore concentrations for each isolate).
  6. Molecular Identification and Phylogenetic Analysis of Selected Actinobacteria (Section 2.6):
    • DNA Extraction Method: The method used for DNA extraction is described briefly, but mentioning any quality control steps (e.g., checking DNA integrity via gel electrophoresis) would add credibility to the methodology.
    • PCR Conditions: The thermocycling conditions for PCR are adequately described, but it might be useful to explain why these conditions were chosen (e.g., optimization experiments or based on previous work). This ensures that readers know the conditions are not arbitrary.
    • Sequence Analysis: The method for assembling and revising electropherograms using SnapGene software is clear, but it would strengthen the section to mention if any statistical tests or thresholds (e.g., sequence identity percentage) were applied when making phylogenetic comparisons.
  7. Statistical Analyses (Section 2.7):
    • Normalization of Data: The equation for transforming data (√(x + 0.5)) is mentioned, but it’s unclear if this transformation was applied to all datasets or only specific ones. Clarifying when and why this transformation was necessary would help future researchers interpret or replicate the data.

For some methods (e.g., culture media, pathogen concentration), it’s unclear why they were chosen. Adding brief justifications for these choices, such as references to preliminary data or published protocols, would enhance the validity and credibility of the study. Methods like DNA extraction, pathogen inoculum preparation, and antibacterial assays could benefit from more discussion on how consistency was ensured across replicates. Including quality control steps would enhance the study's robustness. There is no mention of negative controls for the antibacterial assays or physical treatments. Including or discussing control experiments (e.g., untreated soil samples or blank media) would strengthen the experimental design. Mentioning environmental conditions (temperature, humidity, etc.) during the entire process (from sampling to incubation) would help in ensuring that the study can be replicated under similar conditions.

Author Response

Author Response (Reviewer 2)

Dear Reviewer,

Thank you for your time and kindness in reviewing our manuscript, as well as your valuable suggestions. Below, we provide you with the answer to each of your comments.

Reviewer 2

Comments and Suggestions for Authors

Overall, the paper is scientifically sound, but improvements in clarity, particularly in the discussion and graphical representation, would make the work more accessible and impactful.

The introduction is well-written and contextualizes the problem. However, it could benefit from additional references to recent studies on biocontrol agents or innovative methods for isolating actinobacteria. The transition between problems in chili pepper crops and the use of actinobacteria could be smoother.

Answer: We agree.

It was amended as follows:

In order not to lose sight of the problems of chili cultivation, such as the resistance of Xanthomonas and the search for other alternatives, the paragraph was moved

Lines 47-52: The control of the disease by commercial formulations based on copper has been ineffective due to its excessive use, causing the selection of cooper-resistant Xanthomonas strains [4,5]. Antagonistic microorganisms of the genera Bacillus, Pseudomonas, and Streptomyces, isolated from soil, rhizosphere, phyllosphere, and plant endophytes could be a sustainable alternative or a complement of an integrated management program to control bacterial diseases in plants caused by Xanthomonas spp. [6].

Lines 76-81: Recently, Nguyen et al. [26] demonstrated the potential of Streptomyces sp. JCK-8055 and its antibacterial metabolites in the in vitro and in vivo biocontrol of bacterial wilt and fire blight of apple caused by Erwinia amylovora. Similarly, Padilla-Gálvez et al. [27] reported that Streptomyces sp. TP199 inhibited the in vitro growth of P. carotovorum subsp. carotovorum and P. atrosepticum in tuber slice maceration trials, which reduced maceration halos caused by these phytopathogenic bacteria.

Lines 82-87: On the other hand, pretreatment of soil samples with chemical, physical, or com-bination agents are methods used for selective isolation of actinobacteria [28,29]. Mi-crowave irradiation has been reported as a useful pretreatment in the isolation of ac-tinobacteria by increasing cultured actinobacteria, highlighting that some isolates of the genera Streptomyces, Nocardia, Pseudonocardia, Amycolatopsis and Saccharotrix were only cultured after microwave irradiation [28, 30].

Including a visual representation (flowchart) of the experimental design could improve clarity for readers.

An experimental diagram was included in supplementary material and is included between lines 131-133. The file with this flowchart of the methodology is also added.

The results are presented clearly with adequate statistical support. One suggestion is to provide more detailed comparisons or discussions between the microwave-irradiated and dry-heat-treated samples. Consider expanding on why microwave irradiation yielded more isolates but fewer potent ones.

Answer: We agree. Microwave irradiation was discussed. In addition, the fact that isolates from the irradiated sample will show less antibacterial activity could be due to the isolates (species) themselves (the metabolites it produces, the amount diffused, and the sensitivity of the plant pathogen) than to the effect of microwave irradiation.

It was amended as follows:

  1. Discussion Lines 329-348: The results indicated that microwave irradiation (2450 MHz, 1100 W for 3 min) increased the amount of cultured actinobacteria (CFU g−1 from dry soil) and the number of isolates on ISP2 and ISP3 agar compared to dry heat pretreatment. Wang et al. [30] found that microwave irradiation with 120 W power per 3 min to soil samples increased the amount of culturable actinobacteria in the three media evaluated compared to non-irradiated samples. Similarly, Hamedi et al. [46] reported that the highest percentage of isolates was obtained from microwave-irradiated soil samples, followed by dry heat pretreatment at 120 °C. However, our conditions differ from previous studies in the power used during microwave irradiation, as well as in the temperature of the water prior to irradiation of the soil sample. In our study we found that some actinobacteria of the genera Streptomyces and Amycolatopsis were only culturable after microwave irradiation. Similarly, Wang et al. [30] reported that some isolated Streptomyces were only cultured after microwave irradiation of soil samples. In controlled experiments, Komarova et al. [47] indicated that microwave irradiation (80 W power per 1 min) of soil added with S. Xanthochromogenes increased spore germination with respect to the non-irradiated sample. In another study, Likhacheva et al. [48] found that under controlled conditions microwave irradiation (2450 MHz, 80 W power, 15 to 90 s) suppressed or stimulated the growth and development of Streptomyces spp. In addition, the authors indicated that microwave irradiation as a pretreatment can change the behavior of actinobacteria in the microbiome of soil samples, similar to the results of this study and what was reported by Wang et al. [30].

The discussion is comprehensive but could benefit from further explanation on how the findings compare with other studies in the field. For example, a more in-depth analysis of how these results impact biocontrol practices in agriculture could be included. It would also help to discuss any practical applications or future research directions based on the findings.

Answer: We agree. Future directions of the potential of isolates as biocontrol agents were integrated.

It was amended as follows:

Lines 453-461: To understand how Streptomyces and Amycolatopsis species have the potential to be biocontrol agents beyond their in vitro antibacterial activity, future studies should focus on issues of interaction of chili pepper plants with the inoculation of these actinobacteria or the application of their metabolites when they are infected with Xanthomonas. Other studies are necessary to evaluate their potential use in agriculture, such as determining the concentration, formulation and type of application (foliar or soil) of these actinobacteria (spores or bioactive metabolites) for in vivo control of this foliar disease caused by Xanthomonas spp. with experiments in greenhouse conditions and their validation in field trials.

The figures are useful, but some figures and tables (such as Figure 3) could benefit from clearer labels or more detailed explanations in the captions. There seems to be an issue with the labeling of figures ("two Figure 3" references). Please ensure proper figure numbering and cross-referencing.

Answer: Fixed the error of two Figures 3, verified that the numbers of the figures corresponded due to the addition of a new one and that they would match the text.

The titles of tables and figures were modified as follows:

Lines 264-265: Table 2. Zones of inhibition (inhibition halos) of Xanthomonas sp. BV801 obtained during the antibacterial activity test in PDA by the actinobacteria of the BVEZ and BVEZMW series.

Lines 293-297: Figure 3. Effect of culture medium (PDA, ISP2 and ISP3) on the antibacterial activity of selected actinobacteria of the BVEZ and BVEZMW series against Xanthomonas sp. BV801. The diameter data of the inhibition zone represent the average of four biological repetitions. -: Non-detected. Different letters on each bar for each culture medium indicate significant differences among the media according to Tukey’s (p ≤ 0.05).

Lines 299-301: Figure 4. Representative image of the effect of culture medium (PDA, ISP2 and ISP3) on the antibacterial activity of eight selected actinobacteria from the BVEZ and BVEZMW series against Xanthomonas sp. BV801 by the double layer method of agar.

My special attention was on methodology part:

Sampling of Rhizospheric Soil (Section 2.1):

Clarification on Soil Samples: While the sampling method is generally clear, the description of soil collection (e.g., “Two-soil samples/tree were taken...”) could benefit from more details. Were these samples taken at a specific depth? Was there any particular season or environmental condition considered? These factors can influence microbial populations.

Answer: We agree. The sampling method was described in our previous work where the procedure is described in more detail, so as not to repeat, it was modified as follows:

Lines 93-95: Rhizospheric soil from an avocado (Persea americana Mill.) cv. Hass orchard was collected at the municipality of Ziracuaretiro, Michoacán, México during the dry season in April 2017 as described by Trinidad-Cruz et al. [31].

Homogenization of Subsamples: The text mentions mixing subsamples to obtain a composite sample. It would be helpful to describe how the homogenization process was done to ensure even distribution of microbes (e.g., vortexing or mechanical mixing).

Answer: We agree. For each soil subsample 5 g was weighed, placed together in a sterile polypaper bag and shaken vigorously by hand, it was modified as follows:

Lines 98-99: Before the bacteria isolation procedure, soil subsamples (5 g each) were vigorously mixed inside a sterile bag for 1 min to obtain a composite sample.

Physical Treatments of Composite Soil Sample (Section 2.2):

Heat Treatment Protocol: The protocol for the dry heat method is sufficiently described, but the cooling process after heating is vague (“left to cool at room temperature”). It might be more precise to mention the exact cooling duration or the room temperature range to maintain consistency.

Answer: We agree. The soil sample was allowed to cool at room temperature for 15 min, it was modified as follows:

Lines 104-105: This sample was left to cool at room temperature for 15 min and deposited in a 160-mL milk dilution bottle with 90 mL of sterile distilled water.

Microwave Treatment: The microwave irradiation method needs more clarification on why the specific power and duration (100% power for 3 minutes) were chosen. Was this based on a previous study or preliminary experiments? Additionally, it would be beneficial to discuss potential variations (e.g., microwave type or power output) that could influence reproducibility.

Answer: The methodology used was very similar to that of Wang et al. (2013) but with slight modifications: the selected time (3 min) was taken from Wang et al. (2013), the microwave although they are of different brands both are 2450 MHz, the power used in Wang et al. (2013) was 120 W, while we used 1100 W. Unlike Wang et al. (2013), we placed cold water in the beaker before irradiation. Cooling the water was the result of a preliminary experiment (unpublished data) with another soil sample where the amount of isolates in the irradiated sample was reduced by about half with respect to a sample at 70 °C, this could be due to the fact that the water used in the beaker was not cooled before irradiating. Although unlike the isolates of this study, the irradiated sample showed the actinobacterium with greater antibacterial activity similar to the BVEZ 50 isolate of this study.

Regarding reproducibility, although in Wang et al. (2013) or other works such as Arango et al. (2018) (https://doi.org/10.2174/1874285801812010181) and in our case it was not considered, it would be important to take into consideration both the power, type of microwave or time in subsequent studies irradiating different portions of the same composite sample, because between different soil samples the results vary,  although in general the trend is to increase the number of isolates or the amount of actinobacteria (cfu g−1), as reported by Wang et al. (2013) and Arango et al. (2018).

It was modified as follows by adding the watts of power and MHz of the microwave used:

Lines 109-111: The beaker was placed in the center of the microwave oven (2450 MHz, Panasonic® Inverter System Inside, Osaka, Japan) and irradiated for 3 min (100% Power, 1100 W).

Sterile Controls: Were sterile controls included for both treatments to ensure no contamination was introduced during the physical treatment processes? Mentioning sterile controls for the soil treatments would strengthen the robustness of the methods.

Answer: Once the compound mixture was obtained, sterilized pipette tips were used as spatulas to weigh the 10 g of soil inside sterile 15 mL conical tubes for both composite subsamples (70 °C and irradiated). In addition, the following steps such as mixing the soil with the sterile water from the dilution bottles was performed in a laminar flow hood. What could not be controlled, as in most cases, is the process of weighing the soil samples on the analytical balance. However, it was indicated that the conical tubes were sterilized as follows:

Line 102-107: Composite soil sample was separated in two portions or subsamples. One composite subsample (10 g placed in a sterile 15 mL conical tube) was treated for 1 h with dry heat at 70 °C in a drying oven [32]. This sample was left to cool at room temperature for 15 min and deposited in a 160-mL milk dilution bottle with 90 mL of sterile distilled water. The second composite subsample (10 g) was deposited in a sterile 15-mL conical tube with 4 mL of sterile distilled water and sealed.

Isolation of Actinobacteria (Section 2.3):

Dilution Process: The serial dilution method is commonly used, but the text could be more specific about how the samples were mixed and plated to avoid biases (e.g., vortexing prior to each dilution or using calibrated pipettes).

Answer: We agree. It was amended as follows:

Lines 114-120: Each soil suspension (10−1 dilution) contained in the milk dilution bottles was stirred manually for 5 min and then 1 mL was transferred to a conical tube with 9 mL of sterile water (10−2 dilution) using a micropipette (100-1000 μL). The conical tubes were vortexed for 1 min before continuing with the decimal serial dilutions until the 10−4 dilution. The conical tubes of each dilution were mixed by inversion (five to ten times) and then a 100 μL aliquot was spread with a Digralsky spatula over different culture media with agar in triplicate:

In addition, a figure was added in supplementary material with a diagram of the methodology to improve the understanding of the treatment of soil samples and the isolation of actinobacteria.

Choice of Culture Media: The media used (ISP2 and ISP3) are appropriate, but the rationale for choosing these particular media over others could be explained better. Did preliminary tests show that these media were the most effective for actinobacteria isolation? Were other media tested or considered?

Answer: Although it is true that the culture medium can influence the number of isolates obtained in the same soil sample (https://doi.org/10.1186/s12866-018-1215-7 or https://doi.org/10.3390/microbiolres15020050), but not in others (https://doi.org/10.1038/s41598-020-60968-6). We consider that the variability could depend on both the soil sample, the pretreatment or the culture medium, in addition to the fact that there is no culture medium that favors the isolation of actinobacteria with antibacterial activity. Therefore, ISP2 and ISP3 were chosen, widely used in the morphological characterization of Streptomyces, so they should allow the growth of many species of this type of microorganism. In our case, we found differences in the amount of cultured actinobacteria per gram of soil between culture media for microwave pretreatment, but not for dry heat pretreatment. In addition, using the same culture medium, differences were found between pretreatments.

Other media used in previous studies of the working group were considered, such as potato-dextrose agar (pH 7.0), PDA added with yeast extract pH 7.0, GYM Streptomyces agar, Actinomycete Isolation Agar, among others, but finally we opted for ISP2 and ISP3.

It was amended as follows:

In Materials and Methods, it was indicated that the seeding of the dilutions was carried out in triplicate, the quantification of the CFU g−1 actinobacteria and the statistical analysis.

(2.3) Lines 117-120: The conical tubes of each dilution were mixed by inversion (five to ten times) and then a 100 μL aliquot was spread with a Digralsky spatula over different culture media with agar in triplicate:

(2.3) Lines 125-126: The amount of cultured actinobacteria was quantified from the plate count of dilutions 10−3 or 10−4 and expressed as CFU g−1 of dry soil.

(2.7) Lines 220-222: Student's t-tests of independent samples were used to compare mean bacterial concentration (CFU g−1 of dry soil) between pretreatments and culture media.

In Results:

Lines 236-241: Microwave irradiation increased the concentration of cultured actinobacteria on ISP2 (2.1-fold increase) and ISP3 (3.6-fold increase) agar compared to samples pretreated with dry heat (Figure 1). The counts of culturable actinobacteria were similar for ISP2 and ISP3 agar when the soil sample was pretreated with dry heat, but when irradiated with microwaves the count increased in ISP3 agar (1.6-fold increase) with respect to ISP2 agar (Figure 1).

See Figure 1 (page 6)

Incubation Period: A 14-day incubation period is standard for actinobacteria, but adding details about how the colonies were checked during this period (e.g., intervals for visual inspections) could improve the replicability.

Answer: We agree. It was amended as follows:

Lines 127-129: Petri dishes were checked daily from the first week of incubation, focusing on purification of actinobacteria from dilutions 10−1 and 10−2 and then continued with the remaining dilutions.

Selection of Isolates with Antibacterial Activity (Section 2.4):

Standardization of Pathogen Inoculum: The OD600nm of 1.0 for the Xanthomonas suspension is mentioned, but it’s unclear if this represents the optimal concentration for consistent antibacterial activity measurements. Providing information on how this concentration was determined (e.g., preliminary tests or literature standards) would improve methodological transparency.

Answer: The double layer of agar using an OD600nm of 1.0 (100 μL per 1 mL of medium) is routinely used in the laboratory to isolate and quantify bacteriophages, that concentration of bacteria provides a homogeneous growth of the bacteria in the double layer. We have used it in other experiments and a previous publication with results with little variation (low standard deviation values).

It was amended as follows:

Lines 139-140: Then, strain BV801 was diluted at 1 OD600nm with fresh NYGB to be ready for inhibitory assay as described by Trinidad-Cruz et al. [31].

Antibacterial Activity Assay: The double-layer agar method is well-described, but you could clarify how the measurement of the inhibition zones was standardized. Was the zone measured at the widest point or as an average across different angles? Adding this information will ensure that others can reproduce the results accurately.

Answer: Inhibition halos were measured in most cases where bacterial growth was observed on both sides (Figure a). As it is not expected to obtain isolates that show very large inhibition halos (a small number of isolates), for those cases the inhibition halo was considered from the center of the bite to some nearby growth point of the bacterium that we considered as part of the halo of the isolate as in Figure b to which the color tone was increased to exemplify.

It was amended as follows:

Lines 149-153: Inhibition halos (diameter, inhibition zones= IZ) were measured, considering the center of the agar disc as the central point, towards the margins of Xanthomonas sp. BV801 inhibition. When the inhibition halos were mixed, it was measured from the center of the bite towards the inhibition margin of Xanthomonas sp. BV801 which was considered as part of the inhibition halo of the isolate. Inhibition halos were measured using a digital vernier.

Effect of Culture Media on Antibacterial Activity of Selected Isolates (Section 2.5):

Selection of Culture Media: The rationale behind using PDA, ISP2, and ISP3 media for secondary metabolite production needs to be explained in more depth. Why were these specific media chosen, and how do they compare to other media that might have been used in similar studies?

Answer: Although it is true that in several studies the antibacterial activity is evaluated in the environment where the isolation was carried out, in some cases this is not the case. In our study, since we had isolates in ISP2 and ISP3, we should have done the antibacterial activity (AA) assay for all isolates in both media, but we considered PDA as the first option due to our previous study (in other studies they have used PDA at half its concentration (HPDA) as a medium for the AA assay) where several actinobacteria with AA were observed. The question we asked ourselves was the following: could we have discarded an isolate with activity if ISP2 or ISP3 had been used instead of PDA?, what we did was to test the isolates with the highest inhibition halos in the three culture media, observing that if we chose ISP2, some isolates with good activity would have been discarded in the same way as for ISP3. This could help other studies consider what might be the best means of evaluating AA as a first screening.

It was amended as follows:

Lines 141-142: The AA assay was performed by the double-layer agar according to Salamoni et al. [38], with slight modifications according to Trinidad-Cruz et al. [31].

Replicability of Inoculation: While the inoculation procedure is described well, it might help to include details about how inoculum consistency was maintained across all media types (e.g., consistent spore concentrations for each isolate).

Answer: Due to their morphological characteristics such as mycelium strongly adhered to the environment, waxy, leathery, dusty in appearance, not all of them can show sporulation in the environment from which they were isolated. In liquid medium they tend to form aggregates of different sizes, which makes it difficult to use a common type of inoculum for all isolates and at the same concentration. But it is a good initial test to differentiate between isolates with activity and those that do not, this problem is of many articles, for example, the cross-striation method (there is not a concentration of inoculum the same for all isolates), general inoculum (they are a mixture of mycelium and/or spores adjusted to a McFarland concentration) or agar discs where the microorganism was killed with chloroform vapors (although in this case it depends From the diffusion of the metabolites of the snack to the medium where it was placed, the origin of the bite was the culture of the actinobacteria by striatum or massive striatum where there is also no equal concentration of biomass for all).

While it is true that there will be some variation between the agar discs, we observe with the results of this work that the isolates selected in the first test maintain activity in subsequent tests, it is true that as individual agar discs per plate show more circular halos we could consider in future works to use this procedure as the initial screening.

It is stated that the isolates were grown in their respective culture medium for one week before making the bites (Lines 142-143).

Molecular Identification and Phylogenetic Analysis of Selected Actinobacteria (Section 2.6):

DNA Extraction Method: The method used for DNA extraction is described briefly, but mentioning any quality control steps (e.g., checking DNA integrity via gel electrophoresis) would add credibility to the methodology.

Answer: We agree. DNA integrity was visualized by electrophoresis before proceeding with PCR, but we omitted it because it was possible to proceed with the following steps such as PCR, purification, etc.

It was amended as follows:

Lines 174-176: Genomic DNA integrity was confirmed via 1.2% agarose gel electrophoresis in Tris-acetate-EDTA (TAE) buffer and visualized by ultraviolet fluorescence after staining with GelRed® (Biotium, Inc., Fremont, CA, USA).

PCR Conditions: The thermocycling conditions for PCR are adequately described, but it might be useful to explain why these conditions were chosen (e.g., optimization experiments or based on previous work). This ensures that readers know the conditions are not arbitrary.

Answer: We agree. The temperature and time parameters, except for the alignment temperature of the primers, were taken from the recommendations of the manufacturer of the Taq polymerase. The alignment temperature was a temperature gradient PCR resulting in 58 °C being the optimal alignment temperature for our conditions.

It was amended as follows:

Lines 189-192: The PCR conditions for time and temperature were according to the manufacturer's instructions for the Taq DNA polymerase, except for the alignment temperature. The alignment temperature of the fD1/rD1 primers of 58 °C was obtained from a temperature gradient PCR.

Sequence Analysis: The method for assembling and revising electropherograms using SnapGene software is clear, but it would strengthen the section to mention if any statistical tests or thresholds (e.g., sequence identity percentage) were applied when making phylogenetic comparisons.

Answer: We agree. The criteria that were selected for the sequences were those that showed values greater than 99% similarity of the 16S rRNA gene with the sequences of the isolates.

It was amended as follows:

Lines 205-207: The sequences of the type strains were selected based on the percentage of similarity of the 16S rRNA gene greater than 99%.

Statistical Analyses (Section 2.7):

Normalization of Data: The equation for transforming data (√(x + 0.5)) is mentioned, but it’s unclear if this transformation was applied to all datasets or only specific ones. Clarifying when and why this transformation was necessary would help future researchers interpret or replicate the data.

Answer: We agree. Only one dataset did not meet the assumptions of ANOVA (homoscedasticity and normal distribution) and it was the data that were transformed.

It was amended as follows:

Lines 224-226: The data of the inhibition halos on ISP2 agar were transformed using the following equation:

For some methods (e.g., culture media, pathogen concentration), it’s unclear why they were chosen. Adding brief justifications for these choices, such as references to preliminary data or published protocols, would enhance the validity and credibility of the study. Methods like DNA extraction, pathogen inoculum preparation, and antibacterial assays could benefit from more discussion on how consistency was ensured across replicates. Including quality control steps would enhance the study's robustness. There is no mention of negative controls for the antibacterial assays or physical treatments. Including or discussing control experiments (e.g., untreated soil samples or blank media) would strengthen the experimental design. Mentioning environmental conditions (temperature, humidity, etc.) during the entire process (from sampling to incubation) would help in ensuring that the study can be replicated under similar conditions.

Answer: The answers to the reviewer's doubts have been given in previous points, but we remain pending any other suggestions or questions.

Reviewer 3 Report

Comments and Suggestions for Authors

The paper is an interesting piece of work, particularly with its innovative use of microwave irradiation to enhance the isolation of actinobacteria with antibacterial activity, which led to the discovery of Streptomyces strains with substantial biocontrol potential. Additionally, the research's methodological approach, including 16S rRNA sequence analysis and multiple culture media tests, provides a robust framework for identifying actinobacteria that could be effective against Xanthomonas species, a key agricultural pathogen.

Items to improve:

1) The methodology lacks a detailed justification for some steps, such as why only certain physical treatments (dry heat and microwave) were used. IThe rationale for these choices can be given and compared with other potential methods.

2) The discussion touches on the potential biocontrol capabilities of the isolates but lacks a deeper exploration of the specific mechanisms through which they exert antibacterial activity. More discussion on possible metabolic pathways or secondary metabolites could strengthen the relevance of the findings.

Author Response

Author Response (Reviewer 3)

Dear Reviewer,

Thank you for your time and kindness in reviewing our manuscript, as well as your valuable suggestions. Below, we provide you with the answer to each of your comments.

Reviewer 3

Comments and Suggestions for Authors

The paper is an interesting piece of work, particularly with its innovative use of microwave irradiation to enhance the isolation of actinobacteria with antibacterial activity, which led to the discovery of Streptomyces strains with substantial biocontrol potential. Additionally, the research's methodological approach, including 16S rRNA sequence analysis and multiple culture media tests, provides a robust framework for identifying actinobacteria that could be effective against Xanthomonas species, a key agricultural pathogen.

Items to improve:

1) The methodology lacks a detailed justification for some steps, such as why only certain physical treatments (dry heat and microwave) were used. IThe rationale for these choices can be given and compared with other potential methods.

Answer: We agree. In the introduction, part of the importance of the microwave irradiation method was described and in discussion it was compared with other similar studies with respect to similar or different results. The dry heat method can be used to isolate rare actinobacteria, but normally the temperature is higher, in our case as in other articles is to favor the growth of spores of actinobacteria (generally Streptomyces) that present tolerance to this temperature (70 °C), we think that in the future formulation could generate products based on spores or that adapt to processes such as spray drying.

It was amended as follows:

Introduction

Lines 82-87: On the other hand, pretreatment of soil samples with chemical, physical, or combination agents are methods used for selective isolation of actinobacteria [28,29]. Microwave irradiation has been reported as a useful pretreatment in the isolation of actinobacteria by increasing cultured actinobacteria, highlighting that some isolates of the genera Streptomyces, Nocardia, Pseudonocardia, Amycolatopsis and Saccharotrix were only cultureable after microwave irradiation [28, 30].

2) The discussion touches on the potential biocontrol capabilities of the isolates but lacks a deeper exploration of the specific mechanisms through which they exert antibacterial activity. More discussion on possible metabolic pathways or secondary metabolites could strengthen the relevance of the findings.

Answer: We agree. However, so far we do not have direct data (knowing exactly what metabolites they produce) or indirect data such as sequencing of biosynthetic genes that would allow us to give a possible response to the antibiosis mechanism. Therefore, what we did was to try to find out if there would be reports of the identification of bioactive metabolites from their closest relatives, although this does not mean that our isolates produce them.

Round 2

Reviewer 1 Report

Comments and Suggestions for Authors

The authors have addressed all previous comments effectively, significantly enhancing the manuscript’s clarity and presentation. This version meets the journal's standards, and I recommend it for publication in its current form. 

Reviewer 2 Report

Comments and Suggestions for Authors

/